# High-Dimensional Satellite Image Compositing and Statistics for Enhanced Irrigated Crop Mapping

**Michael J. Wellington *** and **Luigi J. Renzullo**

Fenner School of Environment and Society, The Australian National University, Canberra, ACT 2601, Australia; luigi.renzullo@anu.edu.au
* Correspondence: michael.wellington@anu.edu.au

**Abstract:** Accurate irrigated area maps remain difficult to generate, as smallholder irrigation schemes often escape detection. Efforts to map smallholder irrigation have often relied on complex classification models fitted to temporal image stacks. The use of high-dimensional geometric median composites (geomedians) and high-dimensional statistics of time-series may simplify classification models and enhance accuracy. High-dimensional statistics for temporal variation, such as the spectral median absolute deviation, indicate spectral variability within a period contributing to a geomedian. The Ord River Irrigation Area was used to validate Digital Earth Australia's annual geomedian and temporal variation products. Geomedian composites and the spectral median absolute deviation were then calculated on Sentinel-2 images for three smallholder irrigation schemes in Matabeleland, Zimbabwe, none of which were classified as areas equipped for irrigation in AQUASTAT's Global Map of Irrigated Areas. Supervised random forest classification was applied to all sites. For the three Matabeleland sites, the average Kappa coefficient was 0.87 and overall accuracy was 95.9% on validation data. This compared with 0.12 and 77.2%, respectively, for the Food and Agriculture Organisation's Water Productivity through Open access of Remotely sensed derived data (WaPOR) land use classification map. The spectral median absolute deviation was ranked among the most important variables across all models based on mean decrease in accuracy. Change detection capacity also means the spectral median absolute deviation has some advantages for cropland mapping over indices such as the Normalized Difference Vegetation Index. The method demonstrated shows potential to be deployed across countries and regions where smallholder irrigation schemes account for large proportions of irrigated area.

**Keywords:** geomedian; smallholder; irrigation; random forest; high-dimensional

## 1. Introduction

Mapping and quantifying irrigated areas are critical to local, national, and international organizations that aim to understand and govern land and water resources for food security and sustainable development. National estimates are often based on non-exhaustive on-ground surveys, or low-resolution remote sensing estimates for continental scale applications. This can result in a broad range of estimations for a given area. For example, Vogels et al. [1] recently found that irrigated area across the Horn of Africa was on the order of two to four times greater than several official estimates.

Irrigated farming in Zimbabwe occurs along a spectrum of scales, from small informal plots to large commercial operations. Based on government records, Landsat, and MODerate resolution Imaging Spectroadiomater (MODIS) imagery, current estimates of irrigated area range from 123,900 to 202,600 ha [2,3]. Most of the area included in these estimates is in Mashonaland where large-scale irrigation schemes are prevalent [4]. Irrigated area estimates for Matabeleland, where smallholder schemes predominate, are much lower [4]. However, these estimates overlook some irrigation schemes and smallholder activity. It is hypothesized that official figures underestimate irrigated area, as Vogels et al. [1] observed

for the Horn of Africa. Agencies in Zimbabwe are interested in detailed irrigated cropland mapping for the region of Matabeleland, as this would inform natural resource policies and agricultural research, development, and extension priorities [5]. This region continues to be affected by drought, poverty, and food and water scarcity, so sustainable irrigation is critical to development efforts [6,7]. A reliable method of irrigated cropland classification that accounts for smallholder activity in Matabeleland is therefore required.

Smallholder irrigation activity in tropical environments remains more difficult to accurately map than large-scale commercial irrigation. Small and irregular field size and shape, synchrony of green vegetation phenology in the wet season, cloud cover during crop seasons, and in-field heterogeneity are major challenges for remote sensing of smallholder irrigation [8,9]. The Sentinel-2 satellite mission offers increased sensor resolution capability over the Landsat and MODIS satellites which have been used extensively for land use classification, including irrigated area, over recent decades. Recent work towards improved mapping of small-scale irrigation has largely focused on the application of Sentinel-2 imagery [10].

Several classification techniques have been tested for efficacy on small-scale irrigated areas. Vogels et al. [1] applied a supervised, object-based approach on dry-season mosaic images, with object symmetry and roundness variables included in the random forest classification model. Bousbih et al. [11] explored the fusion of Sentinel-2 optical imagery with a soil moisture product, although the relatively coarse resolution of satellite soil moisture data limits their application in smallholder contexts. Hollander [12] collected training polygons in Mozambique to apply a Sentinel-2 supervised learning model, although found that the ground-collected training data did not include sufficient samples of non-irrigated areas that shared similar spectral characteristics, such as light seasonal vegetation, to fit a sound classification model. It is likely that Vogels et al. [1] approach of collecting training data from visual interpretation produces a better representation of landscape variability, and, therefore, a more robust supervised classification model.

Combining finer resolution satellite imagery with novel approaches to land use classification problems offers a path towards more reliable irrigated area detection. Traditionally, land use classification methods have tended to use a selection of clean images, or simple composite mosaics of clean images [1,13]. The high-dimensional geometric median, hereafter geomedian, has been proposed as a way of constructing high-quality, cloud-free composites whilst preserving high-dimensional relationships between spectral bands [14]. It was developed with the aim of replacing the need for temporal stacks of poorer quality images, which is a popular means of training complex classification models [1,13–17]. Additionally, high-dimensional statistics of temporal variation [18] may also be important predictors of irrigation and other agricultural activity due to the spectral variability associated with cultivation, crop growth, and harvest activities. Therefore, augmenting geomedian images with high-dimensional statistics of time-series may enhance the accuracy of land use classification models for irrigated cropland.

Digital Earth Australia, an Open Data Cube (odc) initiative, offers several high-dimensional statistical products of value for land use classification. For example, an annual geomedian product derived from Landsat-8 is available for the entire Australian continent from 2013 to 2018 [14,19]. Additionally, a triple median absolute deviation [20] product is available for the same period, also derived from Landsat-8 [18,20]. The rationale for this product is change detection and machine learning for land use classification, especially over areas that undergo large changes in cover within a year, such as irrigated croplands [18]. The triple median absolute deviation product has three measures of temporal variation: the Euclidean median absolute deviation, spectral (cosine) median absolute deviation (SMAD), and Bray Curtis dissimilarity. Of these, the SMAD is most appropriate for highlighting areas of change within the period contributing to a given geomedian [18]. Therefore, the SMAD may be an important and useful variable in machine learning models for cropland mapping.



The analysis-ready annual products available from Digital Earth Australia make data acquisition simple, although recalculation of high-dimensional statistics is necessary for applications beyond the Australian continent. For example, high-dimensional products are not yet available in Digital Earth Africa, a parallel odc initiative to Digital Earth Australia. Application of high-dimensional compositing across satellites and continents therefore requires a methodology that calculates necessary images and statistics from available reflectance data.

This research paper aims to demonstrate the improvements in irrigation area mapping resulting from the use of high-dimensional statistics and supervised satellite image classification using odc infrastructure. The paper first demonstrates the use of existing geomedian and SMAD products over a much-studied irrigation scheme in Australia. Secondly, geomedian and SMAD are derived from Sentinel-2 imagery through Digital Earth Africa. Finally, the performance of the high-dimensional dataset approach to classification of smallholder irrigation schemes is evaluated with reference to existing mapping products.

## 2. Materials and Methods

### 2.1. Site Selection and Information

The Ord River Irrigation Area (ORIA) surrounding the town of Kununurra in Western Australia was chosen for validating the Digital Earth Australia products (Table 1). This was because it is a well-studied, irrigated area, and shares similar biophysical and climatic characteristics to irrigation schemes in southern Africa, with frequent cloud cover in the monsoonal wet season [21]. Furthermore, the ORIA supports numerous annual crops in addition to perennial tree crops, despite having less within-field heterogeneity and much larger field size than smallholder irrigation schemes [22].

In total, three sites in Zimbabwe were selected from active irrigation schemes covered by phase 2 of the Transforming Irrigation in Southern Africa (TISA) project, funded by the Australian Centre for International Agricultural Research (ACIAR). The three sites: Silalatshani, Nabusenga, and Lungwalala, are geographically disparate within Matabeleland and vary in scale (Table 1). Furthermore, they are surrounded by various other land types including dryland (rainfed) cultivation, water bodies, and natural vegetation. Locations were confirmed by TISA project leaders [23].

**Table 1.** Irrigation schemes used for irrigated land use classification, their location, coordinates, and approximate area.

| Irrigation Scheme | Location | Coordinates | Area Equipped for Irrigation (ha) |
|---|---|---|---|
| Ord River Irrigation Area | Kununurra, Western Australia | −15.601, 128.762 | 14,000 [24] |
| Silalatshani | Matabeleland South Province, Zimbabwe | −20.799, 29.296 | 442 [25] |
| Nabusenga | Matabeleland North Province, Zimbabwe | −17.462, 28.063 | 19 (measured) |
| Lungwalala | Matabeleland North Province, Zimbabwe | −17.938, 27.561 | 132 (measured) |

### 2.2. Data Collection and Preprocessing-Digital Earth Australia

Data for the ORIA were collected within the Digital Earth Australia 'sandbox', which provides access to odc products in a Jupyter Notebook environment. The Landsat-8 geomedian product (ls8_nbart_geomedian_annual) was generated at 25-m resolution for the year 2017. The triple median absolute deviation product (ls8_nbart_tmad_annual) was derived with the same resolution, extents, and period. Then, three indices were calculated on the geomedian product: Normalized Difference Vegetation Index (NDVI), Normalized Difference Water Index (NDWI), and Bare Soil Index (BSI) using the Digital Earth Australia indices package [26]. The geomedian and triple median absolute deviation datasets were

then merged to form a 12 variable classification dataset comprising six spectral bands, three indices, and three measures of temporal variation (Table 2).

**Table 2.** Variables used for irrigated land use classification for the Ord River Irrigation Area site.

| Group | Variable | Band or Source |
|---|---|---|
| Spectral bands | Blue | B1 |
| | Green | B2 |
| | Red | B3 |
| | Near Infrared | B4 |
| | Shortwave Infrared 1 | B5 |
| | Shortwave Infrared 2 | B6 |
| Indices | Normalized Difference Vegetation Index (NDVI) | [27] |
| | Normalized Difference Water Index (NDWI) | [28] |
| | Bare Soil Index (BSI) | [29] |
| Temporal variation | Spectral median absolute deviation (SMAD) | [18] |
| | Euclidean median absolute deviation (EMAD) | [18] |
| | Bray-Curtis Dissimilarity (bcdev) | [18] |

### 2.3. Data Collection and Preprocessing-Digital Earth Africa

Data for the three Matabeleland sites were generated and collected from the Digital Earth Africa 'sandbox', a parallel initiative to Digital Earth Australia. All available cloud optimized Sentinel-2 (s2_l2a) images were collected in 10-m resolution over each site for the year 2019 [30]. This means that cloudy pixels were attributed as missing values and thus ignored in the calculation of high-dimensional composites and statistics.

The two classification datasets were retrieved for each Matabeleland site: an annual geomedian composite, and a stack of four (January–March, April–June, July–September, October–December) geomedian composites. Cloud effects in wet season months meant that complete geomedian composites could not be generated for each month, so the 3-monthly (quarterly) approach was taken. Annual and quarterly geomedian images were calculated with the odc package using the command 'xr_geomedian' which computes geomedians from a defined image stack [31]. The geomedian (g) was calculated on a collection of images ($x, \ldots, x_n$), based on Roberts et al. [14], as:

$$\text{g} = \underset{x}{argmin} \sum_{i=1}^{n} ||x - x_i|| \tag{1}$$

where *argmin* is the "argument of the minima" [14,18].

The SMAD was calculated on the six spectral bands for Sentinel-2 (Table 3). The SMAD was defined, based on Roberts et al. [18] as:

$$\text{SMAD} = \text{median}(\text{cosdist}(x^{(t)}, g), t = 1, \ldots, n) \tag{2}$$

where x is a temporal stack dataset of images over a given period (t) contributing to the geomedian (g). The cosine distance was calculated as:

$$\text{cosdist}(x, g) = 1 - x^T g / (||x|| \cdot ||g||) \tag{3}$$

where the numerator on the righthand side of Equation (3) is the dot product of spectral data for vectors x and g and $||\cdot||$ is the product of the L-2 norms of each vector.

The SMAD calculation used to collect data for Matabeleland sites was validated against the existing Digital Earth Australia SMAD product over the ORIA before application in Digital Earth Africa. SMAD was chosen over other high-dimensional deviation statistics due to its relative capacity to highlight change within periods of interest [18]. The NDVI, NDWI, and BSI were calculated for each geomedian image with the Digital Earth Africa indices package [32].

The collection process resulted in an annual dataset of 10 variables comprising six spectral bands, three indices, and SMAD for each site (Table 3). Consequently, the stacked classification dataset of four quarterly geomedians comprised 40 (4 quarters per year × 10 variables) classification variables.

**Table 3.** Variables used for irrigated land use classification for three irrigation schemes in Matabeleland.

| Group | Variable | Formula | Band or Source |
|---|---|---|---|
| Spectral bands | Green | | B1 |
| | Red | | B2 |
| | Blue | | B3 |
| | Near Infrared (NIR) | | B4 |
| | Shortwave Infrared 1 (SWIR1) | | B5 |
| | Shortwave Infrared 2 (SWIR2) | | B6 |
| Indices | Normalised Difference Vegetation Index (NDVI) | $(NIR − Red)/(NIR + Red$ | [27] |
| | Normalised Difference Water Index (NDWI) | $(NIR − SWIR)/(NIR + SWIR)$ | [28] |
| | Bare Soil Index (BSI) | $((Red + SWIR) − (NIR + Blue))/((Red + SWIR) + (NIR + Blue))$ | [29] |
| Temporal variation | Spectral median absolute deviation (SMAD) | Equation (2) | [14] |

### 2.4. Data Sampling and Classification

Approximately 100 polygons were drawn over each image and labeled as either 'irrigated' or 'other' based on visual interpretation as shown in Figure 1. The rule for labeling a field as 'irrigated' was the appearance of bright green vegetative crop in at least one of the single-time images within the year of interest. For the ORIA, this was generally in the dry-season when water from channels is applied to crops [22]. For the Matabeleland schemes, each site has been studied as part of the TISA project and are known to be irrigated from channels, especially in the late dry-season. However, visual inspection of the image time-series ensured only actively irrigated fields were included. The polygon shapefiles and geomedian images were then read into R and 80% of polygons were sampled as training polygons, with the remaining 20% retained as validation polygons. The sp package was used to randomly sample 80,000 pixels from within the training polygons and 20,000 pixels from the validation polygons [33]. The caret package was then used to partition the sample into 80% training data and 20% validation data [34]. The classification was therefore pixel-based, although training and validation data were sampled from separate polygons.

The randomForest package was used to train the classification model with 500 trees. Variable importance for the random forest model was reported using mean decrease in accuracy. The caret package was used to generate confusion matrices, overall accuracies, and Kappa coefficients for model performance on the validation data [34,35]. Finally, the relevant classification model was applied to the entire extent for each site to classify pixels not included within either training or validation datasets.

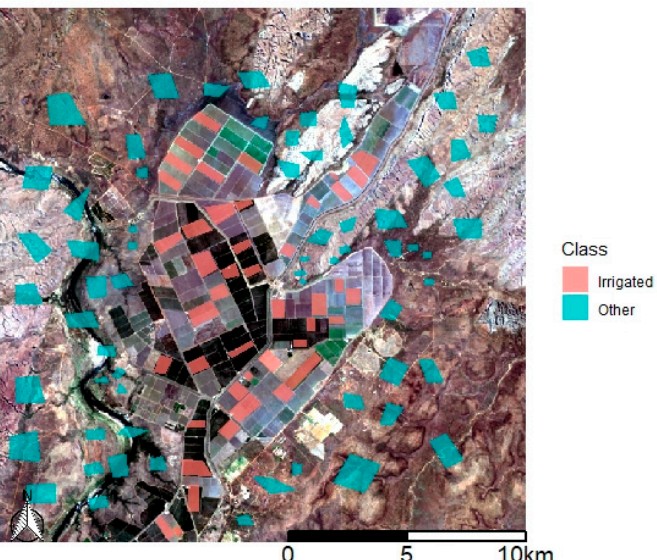

**Figure 1.** Annual geomedian image of the Ord River Irrigation Area overlaid with hand-drawn polygons labeled as either 'Irrigated' or 'Other' by visual interpretation, from which 80,000 training and 20,000 validation pixels were randomly sampled for classification.

*2.5. Comparison to Existing Products*

The Global Map of Irrigated Areas was inspected on the AQUAMAPS web application [36–38]. As none of the Matabeleland sites were classified as equipped for irrigation in the Global Map of Irrigated Areas, no further comparisons were made. For the ORIA, the AQUAMAPS product 'percent of area equipped for irrigation' was downloaded for comparison to classification results.

The FAO portal for monitoring Water Productivity through Open access of Remotely sensed derived data (WaPOR) [39] was used as comparison for the Matabeleland sites. The continental scale, 250-m resolution, WaPOR land use classification product was downloaded from the WaPOR database [40] and cropped to the extent of each site. As the product comprises 24 land use classes, all classes except 'cropland, irrigated' were combined to form an 'other' class for ease of comparison. Confusion matrices were then generated for the WaPOR classification against the validation (20,000 pixels) dataset using the caret package [34]. Accuracy statistics for the WaPOR classification were compared with those for the high-dimensional classification method.

**3. Results**

*3.1. Calculation of Geomedian and Spectral Median Absolute Deviation*

Geomedians and SMAD were derived as existing Landsat-8 based datasets from Digital Earth Australia and recalculated from Sentinel-2 images in the Digital Earth Africa platform. Visual inspection of SMAD plotted as a single band image demonstrates its potential for use in classifying cropland areas, along with other established predictors such as NDVI (Figure 2).

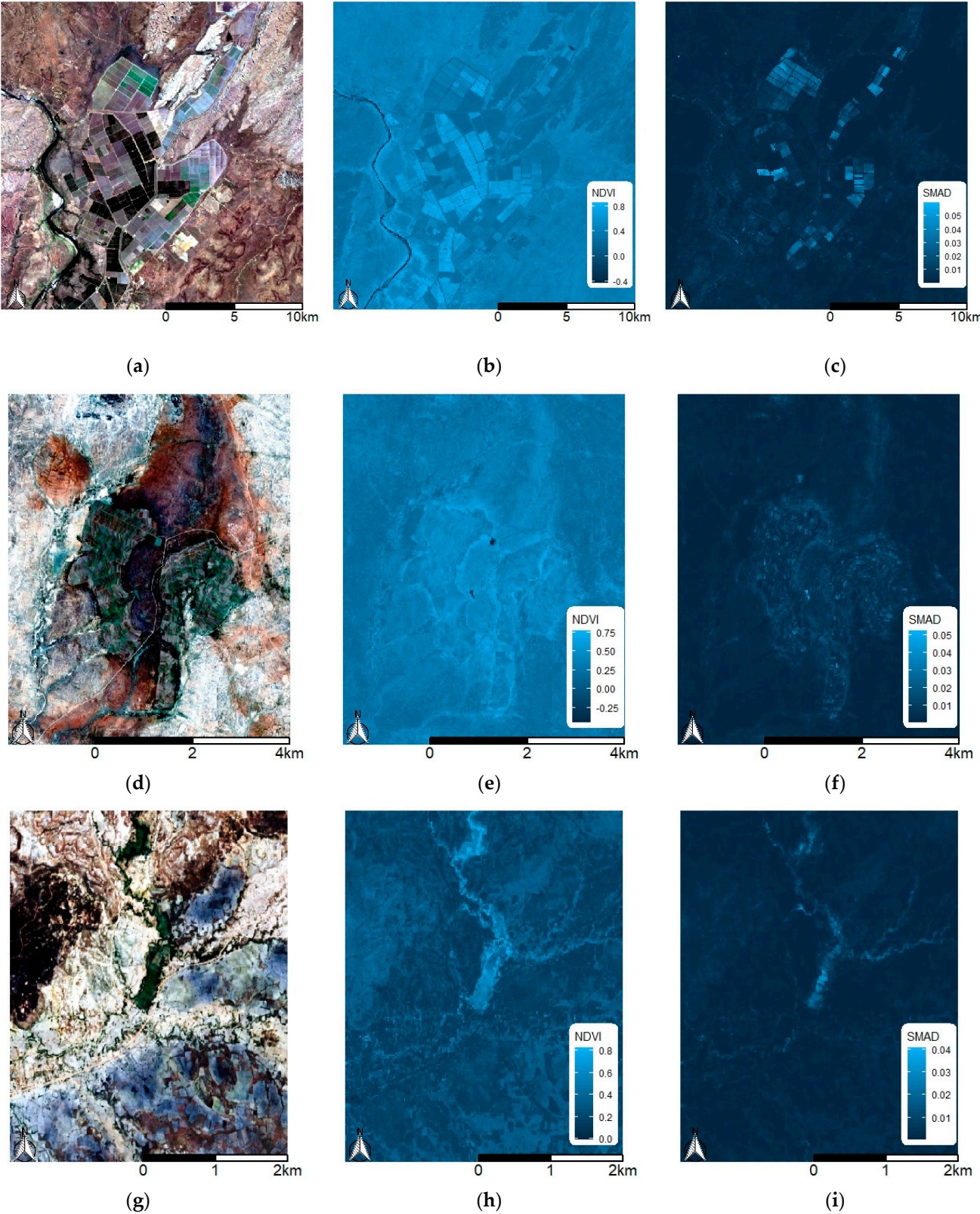

**Figure 2.** *Cont.*

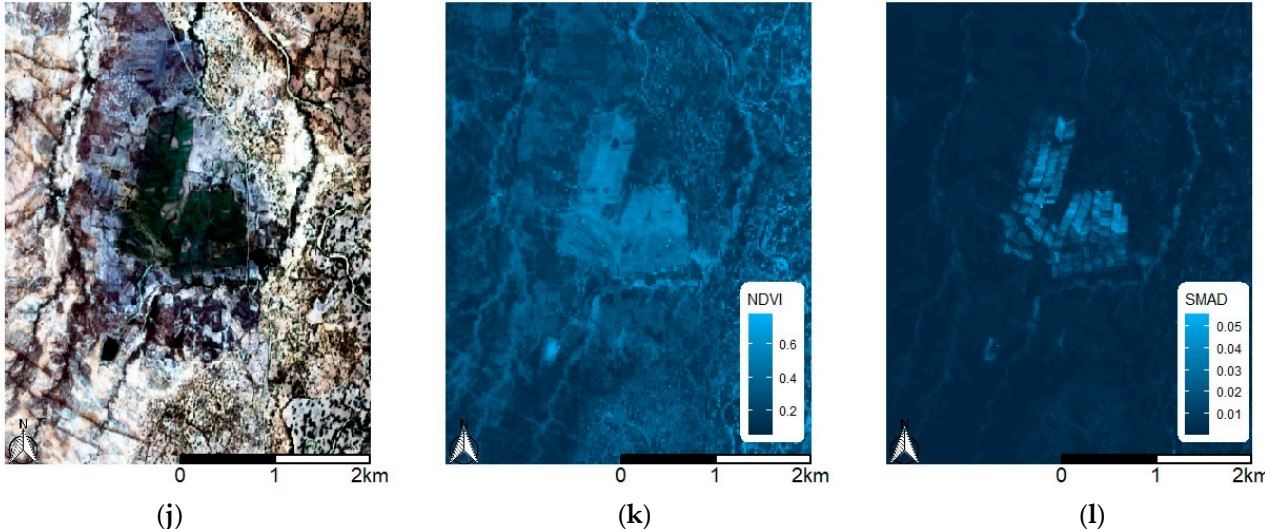

**Figure 2.** Images of the (**a**–**c**) Ord River, (**d**–**f**) Silalatshani, (**g**–**i**) Nabusenga, and (**j**–**l**) Lungwalala irrigation schemes showing (**a**,**d**,**g**,**j**) the calculated geomedian true color composite image, (**b**,**e**,**h**,**k**) a single band image for Normalized Difference Vegetation Index (NDVI) calculated on the geomedian, and (**c**,**f**,**i**,**l**) a single band image for the calculated spectral median absolute deviation (SMAD).

### 3.2. Irrigated Area Classification

Confusion matrices and prediction statistics for each classification model show model performance on validation data (Table 4). All models accurately classified irrigated areas for all sites, with each model giving overall accuracy levels greater than 84% and Kappa coefficients greater than 0.6. The stacked quarter datasets gave better accuracy results than the annual datasets.

**Table 4.** Confusion matrices, overall accuracy, and Kappa coefficients for classification model performance on validation data (20,000 pixels) for all datasets used.

| Ord River Irrigation Area | Observed | |
|---|---|---|
| Predicted | Irrigated | Other |
| Irrigated | 8382 | 351 |
| Other | 2820 | 8447 |
| Overall accuracy (%) | 84.1 | |
| Kappa coefficient | 0.69 | |
| **Silalatshani** | | |
| (A)   Annual dataset | Observed | |
| Predicted | Irrigated | Other |
| Irrigated | 2583 | 226 |
| Other | 1554 | 15,637 |
| Overall accuracy (%) | 91.1 | |
| Kappa coefficient | 0.69 | |
| (B)   Stacked quarter dataset | Observed | |
| Predicted | Irrigated | Other |
| Irrigated | 3498 | 0 |
| Other | 627 | 15,860 |
| Overall accuracy (%) | 96.8 | |
| Kappa coefficient | 0.90 | |

**Table 4.** *Cont.*

| | | | |
|---|---|---|---|
| **Nabusenga** | | | |
| (A) Annual dataset | | Observed | |
| Predicted | | Irrigated | Other |
| Irrigated | | 4284 | 0 |
| Other | | 583 | 15,133 |
| Overall accuracy (%) | | 97.1 | |
| Kappa coefficient | | 0.92 | |
| (B) Stacked quarter dataset | | Observed | |
| Predicted | | Irrigated | Other |
| Irrigated | | 4789 | 0 |
| Other | | 161 | 15,050 |
| Overall accuracy (%) | | 99.2 | |
| Kappa coefficient | | 0.98 | |
| **Lungwalala** | | | |
| (A) Annual dataset | | Observed | |
| Predicted | | Irrigated | Other |
| Irrigated | | 4411 | 46 |
| Other | | 41 | 15,502 |
| Overall accuracy (%) | | 99.6 | |
| Kappa coefficient | | 0.99 | |
| (B) Stacked quarter dataset | | Observed | |
| Predicted | | Irrigated | Other |
| Irrigated | | 4452 | 0 |
| Other | | 0 | 15,548 |
| Overall accuracy (%) | | 1 | |
| Kappa coefficient | | 1 | |

Visual inspection of classification maps (Figure 3) for the annual datasets supports the high accuracy statistics. These plots depict the probability of a given pixel being classified as irrigated, calculated as the proportion of 500 trees in the random forest giving an 'irrigated' vote. Pixels with a probability greater than 0.5 are classified as 'irrigated'. Brighter areas are classified as 'irrigated' with higher model confidence. While pixels with values less than 0.5 are classified as 'other', brightness levels identify some landscape features, such as riparian vegetation, which are prone to misclassification.

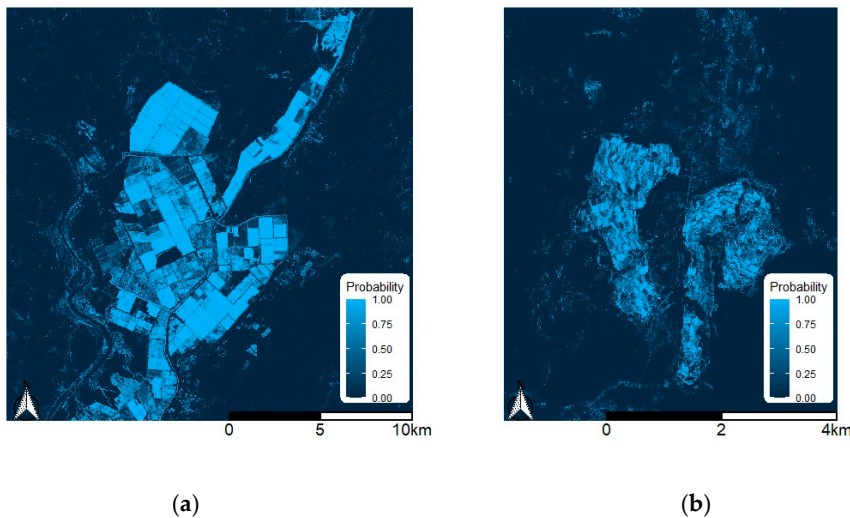

(**a**)                                        (**b**)

**Figure 3.** *Cont.*

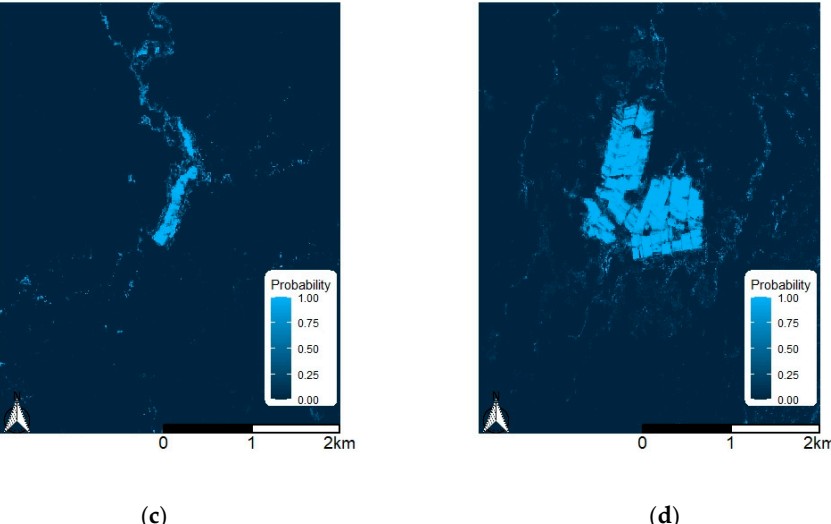

(**c**)          (**d**)

**Figure 3.** Probability of pixels being classified as irrigated, defined as the proportion of 500 decision trees in the random forest model, for annual datasets over (**a**) Ord River Irrigation Area, (**b**) Silalatshani, (**c**) Nabusenga, and (**d**) Lungwalala.

### 3.3. Variable Importance for Irrigated Area Classification

Variable importance analyses for the classification results shown in Table 4 and Figure 3 showed that SMAD was the most important variable for all classification models, except Silalatshani, applied to annual datasets based on mean decrease in accuracy (Figure 4). Higher values indicate a greater decrease in model accuracy if that variable is omitted from the model. The mean decrease in accuracy can therefore be interpreted as a test or summary statistic for variable importance in a random forest classification model.

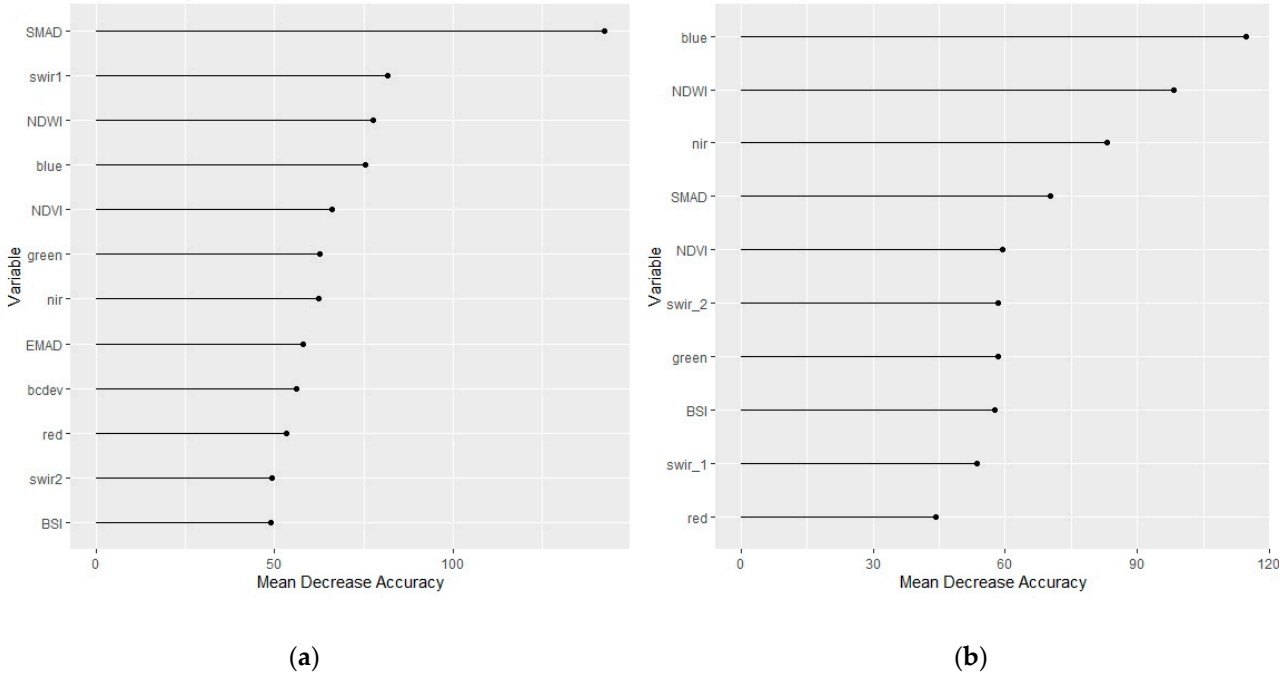

(**a**)          (**b**)

**Figure 4.** *Cont.*

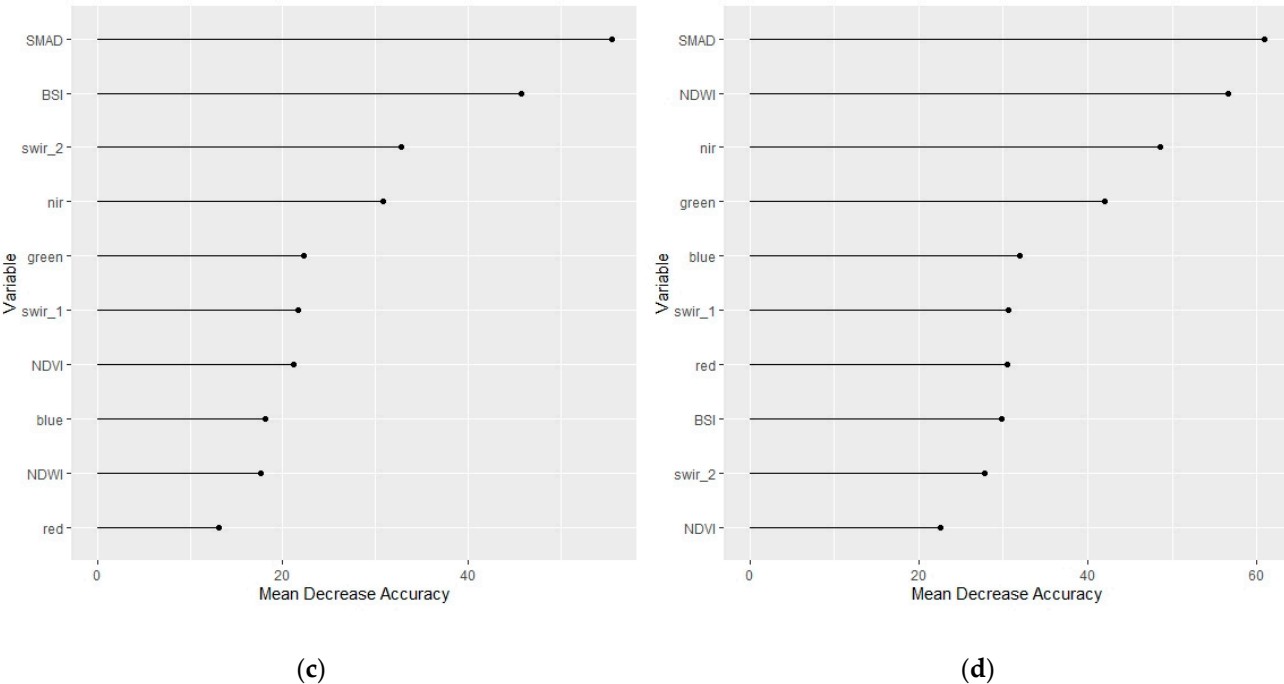

(**c**)                                    (**d**)

**Figure 4.** Variable importance expressed as mean decrease in accuracy for the random forest classification models on annual datasets, shown in Figure 3, for (**a**) Ord River Irrigation Area, (**b**) Silalatshani, (**c**) Nabusenga, and (**d**) Lungwalala.

Figure 5 shows the variable importance plots for the stacked quarterly (4 geomedian images x 10 predictors described in Table 2 = 40 predictors) classification datasets. The SMAD variable for each quarter appeared in the top 15 most important variables of 40 for each site. Like the annual datasets, the importance of indices and spectral bands varied between sites. There was no discernible trend for the importance of specific quarters in any classification model (Figure 5).

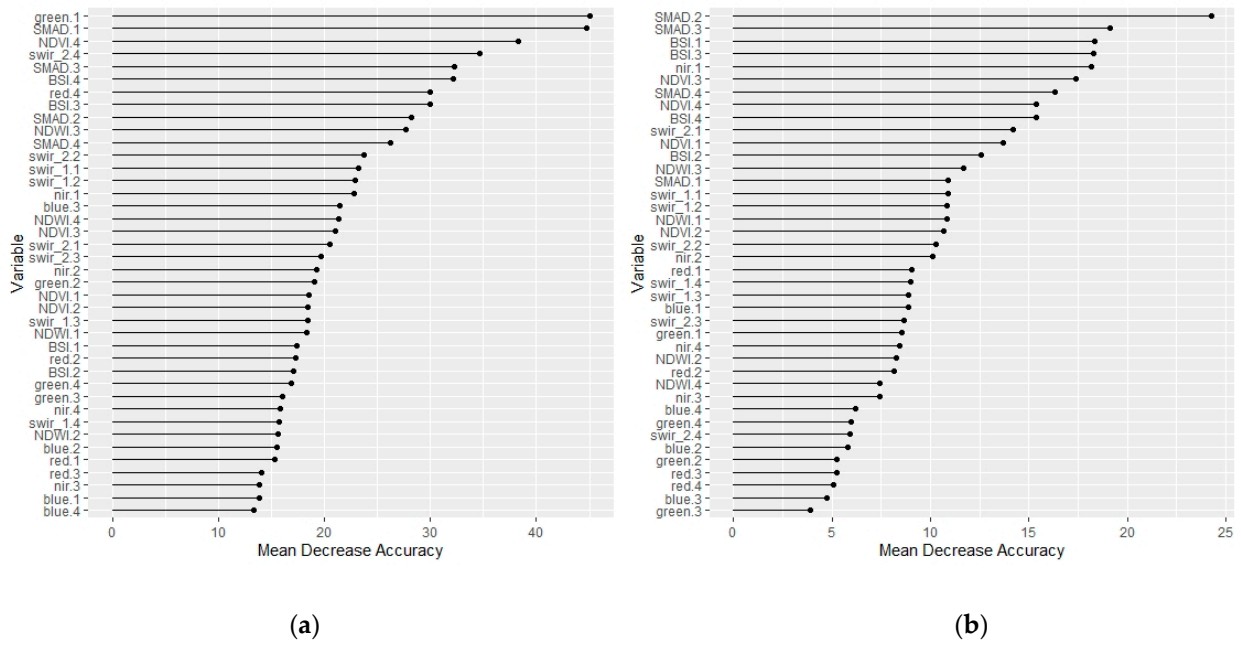

(**a**)                                    (**b**)

**Figure 5.** *Cont.*

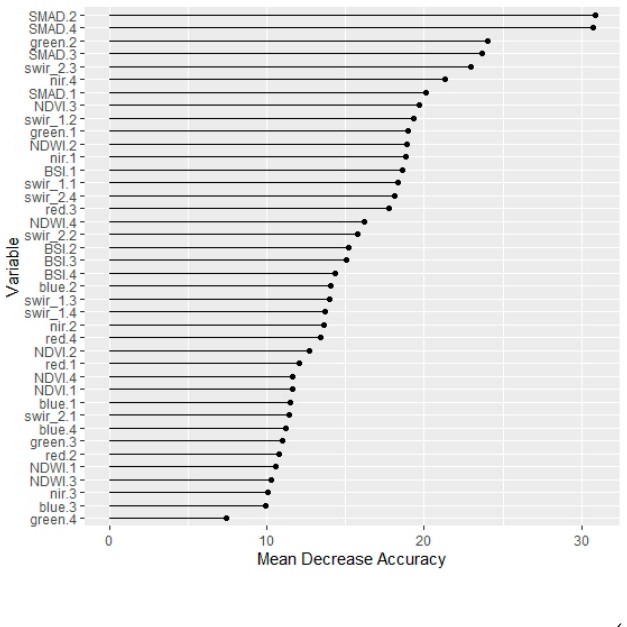

(**c**)

**Figure 5.** Variable importance expressed as mean decrease in accuracy for the stacked quarter datasets where the digit following each variable represents the quarter (1= January–March, 2 = April–June, 3 = July–September, 4 = October–December) for (**a**) Silalatshani, (**b**) Nabusenga, and (**c**) Lungwalala.

### 3.4. Comparison to Existing Products

Comparing the classification results detailed in Table 4 and Figure 3 with existing irrigated land maps illustrates differences in accuracy and resolution. Figure 6 compares the classified image of ORIA predicted on the random forest classification model for the annual Digital Earth Australia products with the Global Map of Irrigated Areas. Notably, this map product classified none of the Matabeleland sites as irrigated. Instead, the Matabeleland sites are compared with the WaPOR land use classification product for the African continent (Figure 6).

The confusion matrices for the WaPOR classification against training and validation data (Table 5) can be compared with those in Table 4. The high-dimensional method on annual datasets produced overall accuracy of 95.9% on average across the three sites, while the average for WaPOR was 77.2%. The average Kappa coefficient of 0.87 was also higher than the WaPOR average of 0.12. The discrepancy between 250-m resolution for the continental WaPOR product and 10-m resolution for the Sentinel-2 dataset contributes to the accuracy results, as illustrated in Figure 6.

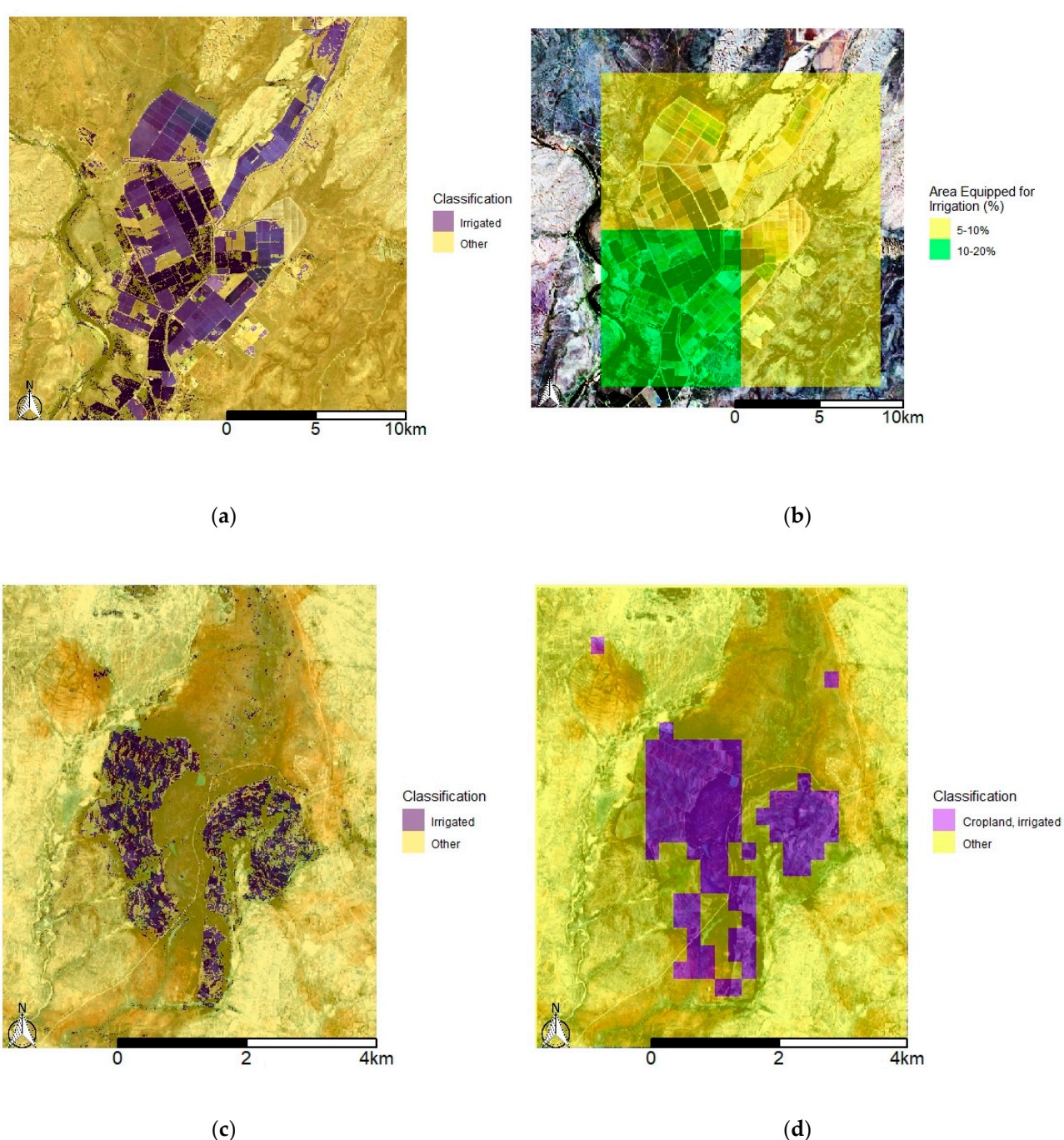

**Figure 6.** *Cont.*

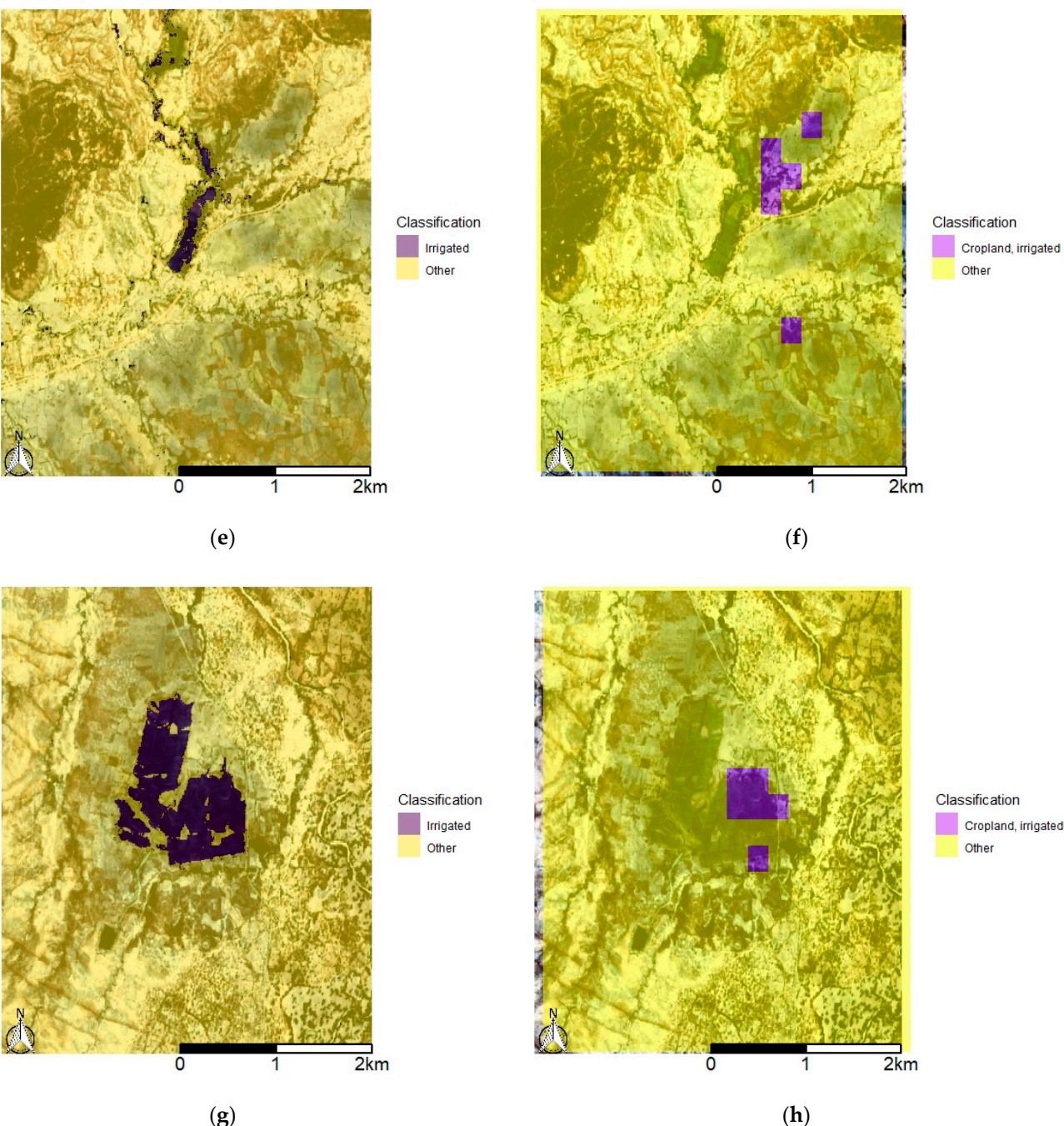

**Figure 6.** True color geomedian composites overlaid with a transparent layer showing classification results for the random forest models trained on annual datasets for (**a**) the Ord River Irrigation Area, compared with (**b**) AQUASTAT's Global Map of Irrigated Areas [37]. Transparent classification layers over true color geomedian composites derived from Sentinel-2 data for (**c**) Silalatshani, (**e**) Nabusenga, and (**g**) Lungwalala are compared with Water Productivity through Open access of Remotely sensed derived data (WaPOR) [40] classification (**d**,**f**,**h**).

**Table 5.** Confusion matrices, overall accuracy, and Kappa coefficients for the Water Productivity through Open access of Remotely sensed derived data (WaPOR) classification product [40] performance on combined training and validation data (20,000 pixels).

| **Silalatshani** | | |
| --- | --- | --- |
| | Observed | |
| Predicted (WaPOR) | Irrigated | Other |
| Irrigated | 489 | 1919 |
| Other | 3644 | 13,948 |
| Overall accuracy (%) | 72.2 | |
| Kappa coefficient | −0.003 | |
| **Nabusenga** | | |
| | Observed | |
| Predicted (WaPOR) | Irrigated | Other |
| Irrigated | 0 | 0 |
| Other | 4867 | 15,133 |
| Overall accuracy (%) | 75.7 | |
| Kappa coefficient | 0 | |
| **Lungwalala** | | |
| | **Observed** | |
| **Predicted (WaPOR)** | Irrigated | Other |
| Irrigated | 1174 | 0 |
| Other | 3278 | 15,548 |
| Overall accuracy (%) | 83.6 | |
| Kappa coefficient | 0.36 | |

## 4. Discussion

### 4.1. High-Dimensional Geomedians and Statistics for Irrigated Cropland Mapping

The collection of geomedian images overcomes the need for temporal stacking of poorer quality, cloud-contaminated images for land use classification. Furthermore, annual geomedian composites are sufficient for cropland mapping as results show negligible improvement in accuracy compared to using a stack of seasonally derived geomedian composites for the given year (Table 4). Marginal improvement in accuracy is likely to be outweighed by the several-fold increase in the number of predictor variables which may lead to model overfitting. Therefore, the existing annual geomedian and SMAD products reduce the dimensionality of classification problem and represent useful data sources for cropland mapping.

The SMAD, as a high-dimensional temporal variation statistic, is critical to the application of annual geomedian composites to cropland mapping. SMAD featured as a key variable to the accuracy of all classification models tested (Figures 4 and 5). Additionally, visual inspection of true color images against the single band image for SMAD in Figure 2 demonstrates that SMAD corresponds strongly to irrigated croplands. Importantly, SMAD also shows greater deviation from surrounding landscapes than the NDVI. Synchrony of crop phenology with surrounding grasslands in the semi-arid tropics is a key limitation to accurate cropland mapping [9,15]. This is especially evident for the Silalatshani site where leakage from irrigation dams and channels [25], and seepage from irrigation plots appears to cause greenness and high NDVI values for surrounding vegetation (Figure 2). The inability to distinguish between rainfed cropland, irrigated cropland, and other green vegetation has been recognized as a limitation of the NDVI [11]. Therefore, the SMAD has desirable properties for irrigated area classification which overcomes some limitations of the NDVI.

Observation of the true color, SMAD, and NDVI plots for the ORIA reveals some further properties of the SMAD. Fields which appear dark green in the true color image show very high NDVI values but are not discernible from the surrounding landscape in the

SMAD plot (Figure 2). Conversely, fields on the north-west corner of the ORIA show low to moderate NDVI values but very high SMAD values. It was hypothesized that high NDVI values and low SMAD values corresponded to fields with perennial tree crops, and that annual cropping was conducted on fields that showed low to moderate NDVI values and high SMAD values. This observation was confirmed by a farmer in the ORIA [41]. Cyclical land cover changes associated with annual crop production give high SMAD values, and fallow periods mean that perennial vegetation may have higher NDVI values in annual geomedian composites than seasonally or annually cropped fields. This means that in addition to cropland mapping, the simultaneous use of SMAD and NDVI may be useful for crop type classification within irrigation schemes.

Beyond SMAD, differences in variable importance between schemes show that indices vary spatiotemporally in their contribution to distinguishing irrigated cropland (Figures 4 and 5). Figure 4 shows that NDWI was an important variable in classifying the ORIA, Silalatshani, and Lungwalala sites but was relatively unimportant for the Nabusenga site. This may be because periodically flood-irrigated fields in the former sites sit within a relatively dry landscape, whereas Nabusenga sits among a wetter landscape meaning NDWI is not an important distinguisher [42]. Additionally, BSI features as an important variable in the Nabusenga classification model, meaning fallow periods in the irrigation scheme may contribute to distinguishing fields from surrounding green vegetation. The modeling results show that indices vary in their relevance to cropland mapping, even within regions and timeframes. Therefore, indices should be selected for irrigated cropland classification with consideration for their relevance to the area of interest.

While all models tested gave very high accuracy statistics, some areas remain prone to confusion and misclassification. The probability of being classified as irrigated maps in Figure 3 show that pixels in the riparian zones of watercourses are subject to misclassification as irrigated. Vegetation in these locations exhibit similar spectral characteristics to irrigated vegetation, given that seasonal waterlogging is likely to occur due to wet and dry extremes of the semi-arid tropics. Noise reduction based on pixel neighborhood information could remove this misclassification.

### 4.2. Application to Irrigated Area Mapping

Generating geomedian composites and high-dimensional statistics shows potential for mapping smallholder irrigation schemes in southern Africa. However, the small scale of these schemes still limits the accuracy of mapping. Figure 6 shows a cleaner classified image for the larger scale ORIA than for any of the Matabeleland schemes, despite the higher resolution of the Matabeleland images. Furthermore, indistinct field boundaries and in-field heterogeneity limit image cleanness for Matabeleland sites [15,25]. Within schemes, there may also be a proportion of fields that are inactive and abandoned at any given time; this proportion ranged from 40 to 60% at Silalatshani from 2013–2018 [43]. These factors contribute to scattering in the classified images shown and continue to limit the accuracy of irrigated cropland mapping across southern Africa. Despite the limitations, the method demonstrated is a substantial advancement on current official irrigated area estimates.

The misclassification of all three Matabeleland sites in the Global Map of Irrigated Areas demonstrates the likely underestimation of official irrigated area statistics in Zimbabwe. Sound classification of the ORIA demonstrates that this product is effective over large-scale schemes in the tropics, but the nature of smallholder Matabeleland schemes means other methods are necessary. Importantly, the AQUASTAT product developers acknowledge this in rating the 'area equipped for irrigation' map quality as 'good' for Australia, and 'poor' for Zimbabwe [38].

The continental-scale WaPOR land use classification product performs more accurately than the Global Map of Irrigated Areas, though may still underestimate irrigated area. However, AQUASTAT is generally referred to for official irrigated area statistics over WaPOR [38]. Our results show WaPOR would provide a more accurate estimate for

Zimbabwe, and likely the African continent, and the demonstrated method would be preferable for regional and country scale mapping.

The demonstrated method has potential to be deployed across the semi-arid tropic areas of sub-Saharan Africa, and other global areas where smallholder irrigation schemes predominate. Geomedians and high-dimensional statistics such as the SMAD can be calculated in Digital Earth Africa or other platforms, although additional computational resources may be required for larger-scale applications. Obtaining training and validation data from additional sites would enhance robustness, as would ground-collected data from within irrigation schemes. However, training data for all other land uses may need to be collected from visual inspection of satellite images due to difficulties obtaining non-biased training data for entire regions using ground surveys [12].

While this method accurately maps smallholder irrigation schemes at known locations, an important limitation is that its ability to detect farmer-led, informal irrigation is unquantified. This form of irrigation generally occurs on dambo landforms; wetlands at the headwaters of river systems where water tables are easily accessible [44]. Field sizes are likely to be smaller than in smallholder schemes and fields are unlikely to be contiguous. Cropping is also likely to be integrated with livestock production and husbandry [45]. These factors combine to make detection of farmer-led irrigation difficult [12]. An extensive ground-collected training dataset for informal irrigation would be required to quantify performance on these areas and further develop the method.

## 5. Conclusions

Annual geomedian composite images combined with high-dimensional statistics of time-series are useful products for irrigated cropland mapping. Digital Earth Australia's geomedian and SMAD products were validated for irrigated cropland classification over the ORIA in north-west Australia. Recalculating annual geomedian composites and the SMAD on Sentinel-2 imagery in Digital Earth Africa generated useful datasets for cropland mapping over Matabeleland, Zimbabwe. Supervised classification using random forest for three pilot sites confirmed that SMAD is a critical variable for irrigated area detection and has advantages over traditionally used vegetation indices such as the NDVI. It may also be useful for differentiating between annual and perennial crops, and detecting cropping activity within a year, season, or other period.

While the Digital Earth Australia analysis-ready products were useful and reduced computation time in this instance, geomedian and high-dimensional statistic calculation packages which allow data collection across continents and satellites may be more valuable. This would negate the need for manual calculation of the SMAD.

The method piloted in this study can be deployed across the entirety of Matabeleland with additional training and validation data. It may also be useful for regional and national mapping in other areas where smallholder irrigation schemes comprise a large portion of irrigated area. Inherent characteristics of smallholder irrigated farming in Zimbabwe continue to limit the accuracy of cropland mapping. However, the application of this method at the regional scale would be an advancement on existing maps and information and has the capacity to reveal previously unrecorded areas of irrigation activity.

**Author Contributions:** Conceptualization and design of the methodology was jointly developed by both authors. L.J.R. conducted supervision and critical review of the formal analysis and writing, and also contributed interpretation of results. M.J.W. conducted data curation, formal analysis, and original draft preparation. All authors have read and agreed to the published version of the manuscript.

**Funding:** The research in this paper was associated with the project 'Transforming Irrigation in Southern Africa' largely funded the Australian Centre for International Agricultural Research under grant number LWR-2016-137.

**Institutional Review Board Statement:** Not applicable.

**Informed Consent Statement:** Not applicable.

**Data Availability Statement:** All data used in this paper is available through Digital Earth Australia, Digital Earth Africa, FAO's WaPOR, and AQUASTAT, code is available at https://github.com/mickwelli/Mapping-smallholder-irrigation.git (accessed on 29 March 2021).

**Acknowledgments:** We acknowledge the generous assistance of Jamie Pittock, Petra Kuhnert, Roger Lawes, Karthikeyan Matheswaran, and Peter Ramshaw in preparing this work. This research was undertaken while supported by the Australian National University (ANU) University Research Scholarship and a Commonwealth Scientific and Industrial Research Organisation (CSIRO) and ANU Digital Agriculture Supplementary Scholarship through the Centre for Entrepreneurial Agri-Technology.

**Conflicts of Interest:** The authors declare no conflict of interest.

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
