# Peer review of "High-Dimensional Satellite Image Compositing and Statistics for Enhanced Irrigated Crop Mapping"

_remotesensing, doi:10.3390/rs13071300_

Round 1

Reviewer 1 Report

This paper should pass review with very few revisions. The authors have done a good job of describing their methods, and offer enough evidence to make their results seem credible. 

My one question is on the detection of irrigated farmland vs. non-irrigated farmland, and how the authors can be sure that the farmland is, in fact, irrigated when applied to other sites where they do not have in situ data. There could be a simple answer for this, for example, all agriculture at the time of year under study depends on water from surface or ground sources. But I'm a little curious about the sensitivity between irrigated vs. non-irrigated cropland. I read the discussion on pages 14-15, but I'm curious as to how irrigated and non-irrigated farmland could be distinguished from each other in the images if that was done by visual interpretation, as the caption on Figure 1 suggests. A little more discussion here, and I think the paper is ready for publication - probably doesn't need another round of review.

Another quibble: I would think it would be informative to apply a random forest trained at one site on the imagery at a second site, in order to gauge the usefulness of automating the method for additional areas. It's up to the authors if they'd like to include such an analysis, but I think it would provide more evidence of the utility of the methods.

Minor Comments:

Typo: Page 1, Line 19: "in the order" should be "on the order"

Sorry, I feel like I'm nitpicking, but this sentence was a little hard to read (Page 1, lines 37-38). At the very least there should be commas in there...
Current estimates of irrigated area range from 37 123,900ha to 202,600ha based on government records and Landsat and MODIS imagery.
Could you change it to something like:
Based on government records, Landsat, and MODIS imagery, current estimates of irrigated area range from 37 123,900ha to 202,600ha.

You might want to cite the authors of the R packages you use.

Reviewer 2 Report

Dear authors, thanks very much for your paper, I see it as very interesting. I have only one problem. Your paper require to know many methods, which are not described in paper and which are not necessary known for readers. I would like suggest to add one chapter, which will describe how to calculate (high-dimensional geometric me-9dian composites (geomedians)). This is not necessary known for all reader.

Then probably will be also good to describe some indexes, which you are using mainly Euclidean median absolute deviation (EMAD) and Bray-Curtis Dissimilarity (bcdev).

You have there link to literature, but explaining this variables will make paper better readable.

Reviewer 3 Report

In this study, the authors present a novel method of using SMAD to map irrigated areas, focusing on smallholders. The manuscript is well-written. The methods and results are presented clearly. I believe the findings from this study will benefit the research community in better mapping irrigation fields for smallholder irrigation schemes dominant region.  

Reviewer 4 Report

This is an interesting paper on improving methods for mapping smallholder irrigation. I think several changes would make it a much stronger paper, however, as I describe below

  1. The accuracy statistics aren’t very convincing since it appears that training and testing points are allowed to come from the same polygons. Can the authors sample training and testing points from different polygons, and make that clear in the paper.
  2. Part of the authors’ argument seems to be that the geomedian and SMAD features are more robust and by implication that it will extrapolate better to areas or years without training. I think this could be true, especially since it doesn’t rely on month-specific phenology details. But they could make this point much stronger if they trained a model in one of their smallholder regions and then tested it on the other two. How much does the accuracy fall if doing that? They could try the various combinations (train on each of the three regions, test on the other two). Those results could help inform how much work needs to be done to train locally or if the features will generalize well.
  3. The paper compares against available irrigation maps such as the WaPOR, which is helpful. But it would be better to also include a reference model that uses the same Landsat or Sentinel-2 data but a different, more traditional set of features, such as the simple quartiles of the annual reflectance and NDVI time series. This would allow readers to understand whether the geomedian and SMAD are a key contribution or whether a RF with simpler features would also work.
  4. L351 – figure 5 does not show this. Do you mean figure 6?
  5. The authors provide an equation for SMAD, which is helpful, but it would be also good to provide an equation or at least a more complete description for the geomedian, as readers should not have to look back at the original paper. Similarly, please provide equations for the 3 VIs used in the study.
  6. I think the paper would also benefit from more description of the image collections used as input to the geomedian and SMAD calculations. They mention ‘cloud optimised Sentinel-2’ does this mean that cloudy pixels are removed prior to computing the geomedian and SMAD?

Round 2

Reviewer 2 Report

Thanks for update. It looks good

Reviewer 4 Report

i think the paper is improved and probably past the point of being publishable. it could have been made stronger if they had been more receptive to the suggestions, in terms of making a convincing case that their approach is better than others. but as it stands it is another approach with some limited amount of evidence.